# FINE-TUNE LANGUAGE MODELS TO APPROXIMATE UN-BIASED IN-CONTEXT LEARNING

## ABSTRACT

In-context learning (ICL) is an astonishing emergent ability of large language models (LLMs). By presenting a prompt that includes multiple input-output pairs as examples and introducing a new query input, models can generate the corresponding output. However, the performance of models heavily relies on the quality of the input prompt when implementing in-context learning. Biased or imbalanced input prompts can significantly degrade the performance of language models. To address this issue, we introduce a reweighted algorithm called RICL (Reweighted In-context Learning). This algorithm fine-tunes language models using an unbiased validation set to determine the optimal weight for each input-output example to approximate unbiased in-context learning. Furthermore, we also introduce a low-cost reweighted algorithm, a linear optimal weight approximation algorithm called LARICL (Linear Approximation of Reweighted In-context Learning). This algorithm requires minimal training cost while providing effective results. We prove the convergence of our algorithm and validate its performance through experiments conducted on a numerical dataset. The experimental findings reveal a substantial improvement in comparison to benchmarks including the performance of casual prompt-based in-context learning and the performance of a classic fine-tuning method.

## 1 INTRODUCTION

Large language models have revolutionized the NLP field across various NLP tasks, including text generation, code generation, and machine translation Hendy et al. (2023); Peng et al. (2023); Jiao et al. (2023); Celikyilmaz et al. (2020); Liu et al. (2023c); Gatt & Krahmer (2018); Zheng et al. (2023b); Perez et al. (2021); Shojaee et al. (2023); Li et al. (2023b); Wang et al. (2023a). A remarkable phenomenon observed in large neural sequence models is in-context learning (ICL), which allows models to learn to make accurate predictions ($f(x_{\text{query}})$) on new inputs ($x_{\text{query}}$) by mapping sequences of $(x, f(x))$ pairs. This behavior, considered a meta-learning ability, has been observed not only in models trained on few-shot learning problems but also in large language models trained on diverse open-domain text Brown et al. (2020); Wei et al. (2022a); Bubeck et al. (2023); Webb et al. (2023); Anil et al. (2023); Bubeck et al. (2023); Liu et al. (2023b). ICL enables neural networks to implicitly learn a mapping from in-context examples to a predictor without requiring updates to the model's parameters Garg et al. (2022); Akyürek et al. (2022); Von Oswald et al. (2023); Liu et al. (2023a). The chain-of-thought prompting method and subsequent related work have highlighted the significance of ICL in enhancing the reasoning ability of large language models Wang et al. (2022); Wei et al. (2022b); Zheng et al. (2023a); Yao et al. (2023).

However, ICL is an input-driven learning mechanism that is extremely susceptible to the influence of historical inputs. Previous studies have discovered that large language models may generate, biased content, and toxic content in a specific context, this issue becomes more severe when the input includes harmful and misleading contexts Ji et al. (2023); Rawte et al. (2023); McKenna et al. (2023); Zhang et al. (2023a); Agrawal et al. (2023). Also, the performance of in-context learning may be harmed because of imbalanced or noisy input-output examples in prompt Brown et al. (2020); Zhang et al. (2021); Min et al. (2021); Liu et al. (2021a). With the widespread adoption of large language models and their availability as services, the need to prevent malicious guidance from generating harmful content has become increasingly critical.

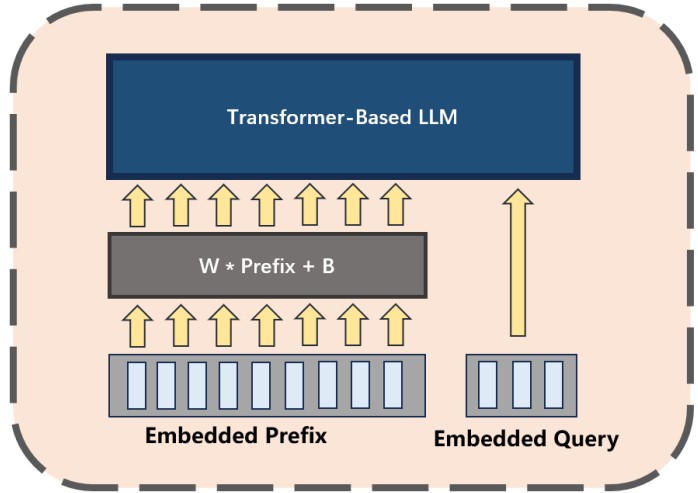

Figure 1: RICL fine-tunes language models by introducing two extra parameters $W$ and $B$ after the embedding layer, in which we transform embedded prefix (prefix with several input-output examples) to $W \cdot \text{Prefix} + B$ to approximate an unbiased in-context. Then we concatenate the reweighted prefix an embedded query as a desired input for the corresponding output.

In this paper, we study the implicit parameter learning mechanism of in-context learning from the perspective of softmax regression (softmax regression has been shown to be equivalent to a distilled model of transformer training) Deng et al. (2023); Gao et al. (2023). Following Theorem 3.4, we suggest: given a input prompt with $m$ input-output pair examples $\{(A_1, b_1), (A_2, b_2), ..., (A_m, b_m), (A_{\text{query}})\}$, language models implement in-context learning to output $(\widetilde{b}_{\text{query}})$ that satisfies

$$\|\widetilde{b}_{\text{query}} - b_{\text{query}}\|_2 \leq O(\exp(R^2 + \log n))\|x_{m+1} - x_m\|_2$$

where

- $x_m = \min_{x \in \mathbb{R}^d} \sum_{i=1}^{m} \|f_i(x_m) - b_i\|_2^2$
- $x_{m+1} = \min_{x \in \mathbb{R}^d} (\sum_{i=1}^{m} \|f_i(x_{m+1}) - b_i\|_2^2 + \|f_{\text{query}(x_{m+1})} - b_{\text{query}}\|_2^2)$
- $f(x) = \langle \exp(Ax), \mathbf{1}_n \rangle^{-1} \exp(Ax)$

To address the challenge of imbalanced, misleading, and noisy input-output examples within the input context, we propose leveraging the approach of reweighting input vectors as an approximation to achieve unbiased in-context learning. Building upon the work presented in Ren et al. (2018), this method involves assigning appropriate weights to the input vectors, allowing us to mitigate the impact of imbalances and biases present in the prefix. By reweighting the input vectors, we aim to create a more accurate and representative learning process that is less influenced by misleading or noisy examples. This approach provides a practical solution for improving the reliability and fairness of in-context learning in scenarios where the input context may contain imbalanced or misleading information.

In summary, we have made the following contributions

1. We propose the application of a reweighting method on embedding vectors of prompts to enable unbiased in-context learning. As shown in Figure 1, we introduce Algorithm 1 (RICL) that outlines the process of learning with reweighted examples. To address the high cost associated with fine-tuning Language Models, we also provide Algorithm 2 (LARICL), which is easy to implement and has a minimal training cost.

2. We establish the Lipschitz-smooth property for the gradient of the training objective and derive an upper bound for it. This allows us to strictly prove the convergence of our proposed algorithms, providing a strong theoretical foundation for our work.

3. Through extensive experiments, we validate the effectiveness of our methods. Our proposed RICL approach outperforms significant improvement by comparing with the fine-tuning

method and casual prompt-based in-context learning method on numerical datasets. Besides, we show the robustness of RICL through experiments on prefixes with different extreme distributions.

## 2 RELATED WORK

This section briefly reviews the related research work on In-context learning, Parameter-efficient-fine-tuning (PEFT) algorithm for LLM, and Imbalanced learning. These topics have a close connection to our work.

**In-context learning.** Since the discovery of in-context learning as an emergence ability of LLM Brown et al. (2020), prior works have studied in understanding and improving this ability Liu et al. (2021a); Min et al. (2021); Zhao et al. (2021); Garg et al. (2022); Akyürek et al. (2022); Von Oswald et al. (2023); Zhang et al. (2023b); Ding et al. (2023); Li & Qiu (2023); Wei et al. (2023); Wies et al. (2023); Sun et al. (2023); Meade et al. (2023); Li et al. (2023a); Ye et al. (2023); Ramos et al. (2023). Garg et al. (2022) explores the concept of in-context learning, demonstrating that standard Transformers can be trained to effectively learn linear functions and even more complex function classes, achieving comparable performance to traditional estimators and neural networks. Akyürek et al. (2022) investigates the hypothesis that transformer-based in-context learners implicitly implement standard learning algorithms by encoding smaller models in their activations and updating them as new examples appear, providing evidence through linear regression experiments. Ding et al. (2023) provides a theoretical analysis and empirical experiments demonstrating that prefix language models (prefixLM) outperform causal language models (causalLM) in in-context learning tasks, showing that prefixLM converges to the optimal solution of linear regression while causalLM follows the dynamics of online gradient descent, which is not guaranteed to be optimal.

**Parameter-efficient-fine-tuning (PEFT) algorithm for LLM.** Parameter-efficient-fine-tuning (PEFT) methods enable efficient adaptation of pre-trained language models (PLMs) to various downstream applications without fine-tuning all the model's parameters. Fine-tuning large-scale PLMs is often prohibitively costly. In this regard, PEFT methods only fine-tune a small number of (extra) model parameters, thereby greatly decreasing the computational and storage costs. Recent State-of-the-Art PEFT techniques achieve performance comparable to that of full fine-tuning Hu et al. (2021); Lester et al. (2021); Li & Liang (2021); Liu et al. (2021b); Vu et al. (2021); Ding et al. (2022); Liu et al. (2022); Asai et al. (2022); Liu et al. (2023b); Wang et al. (2023b); Meng et al. (2023). Low-Rank Adaptation (LoRA) Hu et al. (2021), a method that reduces the number of trainable parameters in large-scale pre-trained language models by injecting trainable rank decomposition matrices, achieving comparable or better model quality than full fine-tuning while significantly reducing GPU memory requirements and training throughput. Li & Liang (2021) introduces prefix-tuning, which optimizes a small task-specific vector while keeping the language model parameters frozen, achieving comparable performance with significantly fewer parameters and better generalization to unseen topics. Liu et al. (2022) introduces a new PEFT method called $(IA)^3$ and a task-agnostic recipe called T-Few that achieves super-human performance on unseen tasks. Wang et al. (2023b) introduces multitask prompt tuning (MPT), a method that distills knowledge from multiple task-specific prompts to learn a transferable prompt vector, and then efficiently adapts it to downstream tasks using low-rank updates, outperforming existing methods and even the full fine-tuning baseline with significantly fewer task-specific parameters.

**Imbalanced learning.** Class-imbalance (also known as the long-tail problem) is the fact that the classes are not represented equally in a classification problem, which is quite common in practice. For instance, fraud detection, prediction of rare adverse drug reactions, and prediction gene families. Failure to account for the class imbalance often causes inaccurate and decreased predictive performance of many classification algorithms. Imbalanced learning aims to tackle the class imbalance problem to learn an unbiased model from imbalanced data Han et al. (2005); He & Garcia (2009); Krawczyk (2016); Haixiang et al. (2017); Lin et al. (2017); Wang et al. (2017); Ren et al. (2018); Shu et al. (2019); Hu et al. (2019); Lee et al. (2019); Liu et al. (2020); Zhang et al. (2023c); Xie et al. (2023); Santos et al. (2022). Ren et al. (2018) introduces a novel meta-learning algorithm that assigns example weights based on gradient directions, without requiring additional hyperparameter tuning, achieving impressive performance on class imbalance and corrupted label

problems with limited clean validation data. Liu et al. (2020) presents MESA, a novel ensemble imbalanced learning framework that adaptively resamples the training set to optimize the final metric, providing a robust and transferable solution without relying on heuristic assumptions or high computational cost.

# 3 BACKGROUND: IMPLEMENTING IN-CONTEXT LEARNING IN ATTENTION SCHEME

## 3.1 SOFTMAX REGRESSION

The Transformer model Vaswani et al. (2017), is currently considered the state-of-the-art foundation for large language models. It is designed to map a sequence of input vectors, denoted as $X = [x_1, ..., x_n]$, to a corresponding sequence of output vectors, denoted as $Y = [y_1, ..., y_n]$. In each layer of the Transformer, a matrix $X_l$ (representing a sequence of vectors) is transformed into a new sequence $X^{(l+1)}$. Here, we are interested in auto-regressive (or "decoder-only") transformer models in which each layer computes a self-attention, there is:

$$\text{Attention}(X_l) = D^{-1} \exp(X_l Q K^\top X_l^\top) X_l V$$

where $Q \in \mathbb{R}^{d \times d}, K \in \mathbb{R}^{d \times d}, V \in \mathbb{R}^{d \times d}$ are trainable matrix parameters and $D := \text{diag}(\exp(X_l Q K^\top X_l^\top) \mathbf{1}_n)$. Then each row of $D^{-1} \exp()$ is a softmax.

In contrast to previous studies such as Garg et al. (2022); Akyürek et al. (2022); Von Oswald et al. (2023); Zhang et al. (2023b); Ding et al. (2023), which focused on analyzing the in-context learning phenomenon in the context of linear self-attention (LSA) with linear tasks, our research investigates the in-context learning phenomenon in the context of softmax self-attention (SSA). Building upon the work of Gao et al. (2023), we delve into the training process of the attention mechanism and decompose it to Softmax Regression Deng et al. (2023).

Specifically, we examine the training procedure for a particular layer ($l$-th layer) and single-headed attention, where we have an input matrix denoted as $X_l$ and a target matrix denoted as $X_{l+1}$. Our objective is to minimize the loss function through back-propagation in the training process. We train a back-propagation step to minimize

$$\|D^{-1} \exp(X_l Q K^\top X_l^\top) X_l V - X_{l+1}\|_F^2.$$

We first optimize matrix $V$ that

$$V = V - \eta \frac{\mathrm{d}}{\mathrm{d}V} \|D^{-1} \exp(X_l Q K^\top X_l^\top) X_l V - X_{l+1}\|_F^2$$

where $\eta$ is the learning rate. After that, our training objective becomes

$$\|D^{-1} \exp(X_l Q K^\top X_l^\top) - X_{l+1}(X_l V)^\dagger\|_F^2 \tag{1}$$

where $^\dagger$ is the Moore-penrose pseudo-inverse. Now we define $X := Q K^\top$ and $B := X_{l+1}(X_l V)^\dagger$, $A := X_l \otimes X_{l+1}$. To further simplify the analysis, we narrow our focus to a specific column of the expression $D^{-1} \exp(X_l A X_l^\top)$. In this context, we provide our definition of Softmax Regression:

**Definition 3.1** (Softmax Regression, Deng et al. (2023)). *We define Softmax Regression as follows*

$$f(x) := \langle \exp(Ax), \mathbf{1}_n \rangle^{-1} \exp(Ax)$$

*where $x$ is a column of $X$.*

Also, we provide our definition of Softmax Regression Loss:

**Definition 3.2** (Softmax Regression Loss, Deng et al. (2023)). *We define loss of Softmax Regression as follows*

$$\ell(x) := 0.5 \langle c(x), c(x) \rangle$$

*where $c(x) = f(x) - b$, $b$ is a column of $B$.*

Since Claim 3.3, optimizing $\ell(x)$ is equivalent to optimizing Eq. (1):

**Claim 3.3** (Part 2 and Part 3 of Claim 5.17 in Gao et al. (2023))**.** *We have*

- $\min_{x \in \mathbb{R}^d} \|A_1 X A_2^\top - B\|_F^2$ *is equivalent to* $\min_{x \in \mathbb{R}^d} \|(A_1 \otimes A_2)x - b\|_2^2$

- $\min_{x \in \mathbb{R}^d} \| \exp(A_1 X A_2^\top) - B\|_F^2$ *is equivalent to* $\min_{x \in \mathbb{R}^d} \| \exp((A_1 \otimes A_2)x) - b\|_2^2$

*where* $A_1 := X_l$ *and* $A_2 := X_l$.

### 3.2 Language Models Learn In-context by Optimizing Softmax Regression

In prior works like Brown et al. (2020); Garg et al. (2022); Akyürek et al. (2022); Von Oswald et al. (2023); Liu et al. (2023a), a formal framework for in-context learning has been established. In-context learning refers to the capability of models to generate predictions that are contextual in nature during the inference phase. This implies that when presented with a sequence containing input-label pairs and a query input $P = \{(X_1, y_1), (X_2, y_2), ..., (X_m, y_m), X_{\text{query}}\}$ consisting of $m$ examples, the objective of the model is to predict the label $y_{\text{query}}$ for $x_{\text{query}}$ by leveraging the contextual information provided by the context examples $(X_1, y_1), (X_2, y_2), ..., (X_m, y_m)$. Importantly, the model accomplishes this without modifying its parameters.

In Gao et al. (2023), they propose that softmax self-attention achieves in-context learning through the utilization of Softmax Regression. This can be described as follows:

**Theorem 3.4** (Learning in-context for Normalized Version, Part 1 of Theorem 2.1 in Gao et al. (2023))**.** *Denote in-context learning implicit parameter* $x_t \in \mathbb{R}^d$ *is driven by input prompt* $\text{prefix} = \{(A_1, b_1), (A_2, b_2), ..., (A_t, b_t)\}$. $x_t \in \mathbb{R}^d$ *and* $x_{t+1} \in \mathbb{R}^d$ *satisfy* $x_t \leq R$ *and* $x_{t+1} \leq R$. *Let* $M := O(\exp(R^2 + \log n))$.

*By transitioning* $x_t$ *to* $x_{t+1}$, *we are effectively addressing a fresh normalized softmax regression problem involving*

$$\min_{x \in \mathbb{R}^d} \|f(x) - \widetilde{b}\|_2^2$$

*where* $\widetilde{b} \in \mathbb{R}^d$ *satisfies*

$$\|b - \widetilde{b}\|_2 \leq M \cdot \|x_{t+1} - x_t\|_2$$

Theorem 3.4 establishes that in-context learning achieves implicit parameter optimization by minimizing the Softmax Regression loss. In practical terms, this means that during the inference process, the parameters of the in-context learning mechanism are adjusted in a manner that minimizes the Softmax Regression loss function.

Now, we present our formal definition of the in-context learning loss within the softmax attention scheme:

**Definition 3.5** (In-context learning loss in softmax attention scheme)**.** *Suppose there is an input prompt* $P = \{(A_1, b_1), (A_2, b_2), ..., (A_m, b_m)\}$ *with* $m$ *data points* $(A_i, b_i) \in \mathbb{R}^{n \times d} \times \mathbb{R}^n$ *for all* $i \in [m]$. *We define in-context learning loss as follows*

$$\mathcal{L} := \sum_{i=1}^m L(x, A_i, b_i)$$

*where* $L(x, A_i, b_i)$ *is single in-context learning loss such that* $L(x, A_i, b_i) := 0.5\|f_i(x) - b_i\|_2^2$.

## 4 Reweight In-context Learning Prefix

### 4.1 Learning to Reweight Examples for Robust In-context Learning

Indeed, while in-context learning showcases remarkable meta-learning capabilities by utilizing multiple examples, it also carries potential risks. These risks manifest in scenarios such as biased, noisy, or misleading examples within the given prompt (commonly referred to as noisy label problems) or when the prompt itself is imbalanced and fails to capture the complete distribution domain. Both

noisy label problems and class imbalance issues can adversely impact the performance of the model. In the former case, examples with smaller training losses are preferred as they are more likely to represent clean input-output pairs, whereas in the latter case, algorithms like hard negative mining Malisiewicz et al. (2011) prioritize examples with higher training losses, as they are often associated with the minority class.

In this paper, we draw inspiration from Ren et al. (2018) and propose the application of a reweighted method to address these challenges when implementing in-context learning models. Initially, given an input prefix prefix $= \{(A_1, b_1), (A_2, b_2), ..., (A_m, b_m)\}$ consisting of $m$ examples, we initialize a vector $w = [w_1, w_2, ..., w_m]$ randomly. Consequently, we introduce a reweighted in-context learner, denoted as $\text{ICL}(A, x^*) = f(x^*)$, where $x^* = \arg\min_x \sum_{i=1}^{m} w_i \cdot L(x, A_i, b_i)$ and $L(x, A_i, b_i) = 0.5 \|f_i(x) - b_i\|_2^2$. Here, $f_i(x)$ represents the output of the in-context learner when provided with input $x$ and parameter pair $(A_i, b_i)$.

To approximate the desirable values for $\epsilon$, we train ICL on a small unbiased and clean validation set denoted as $\mathcal{V} = \{(A_i^v, b_i^v) \mid 1 \le i \le |\mathcal{V}|\}$. The validation set $\mathcal{V}$ is distinct from $P$ and satisfies $\mathcal{V} \cap P = \varnothing$. We define our training objective on $\mathcal{V}$ as follows:

**Definition 4.1** (Training objective of reweighted algorithm). *Suppose there is an unbiased validation set $\mathcal{V} = \{(A_i^v, b_i^v), A_i^v \in \mathbb{R}^{n \times d}, b_i^v \in \mathbb{R}^d, 1 \le i \le |\mathcal{V}|,\}$ with its size $|\mathcal{V}|$. Denote $\text{ICL}(A, x^*) = \langle \exp(Ax^*), \mathbf{1}_n \rangle^{-1} \exp(Ax^*)$. Denote $x^* = \arg\min_x = \sum_{i=1}^{m} w_i \cdot L(x, A_i, b_i) = 0.5 \sum_{i=1}^{m} w_i \cdot \|f_i(x) - b_i\|_2^2$. For any metric function L, we define in-context learning performance loss on $\mathcal{V}$ as follows*

$$\mathcal{L}_{valid}(w) = \sum_{i=1}^{|\mathcal{V}|} L(\text{ICL}(A_i^v, x^*), b_i^v)$$

So far, we can compute optimal weight $w^*$ for reweighted in-context learning by optimizing $\mathcal{L}_{\text{valid}}$.

## 4.2 Applying Reweight Method on Transformer

In lathe st section, we introduce the reweight prompt method to protect in-context learning from toxic, biased, noisy prompts. In this section, we implement Definition 4.1 in the transformer structure by reweighting the embedded prompt. We consider a situation as follows: Given $m$ embedded input-output pairs $(A_i, b_i)$, $A_i \in \mathbb{R}^{n \times d}$ and $b_i \in \mathbb{R}^d$ for $i \in [m]$. $n$ represents the number of tokens of input for each input-output example in in-context learning, and $d$ represents the dimension of the model. We combine all pairs as a prefix of input of language model, we denote it as prefix $:= [A_1, b_1^\top, A_2, b_2^\top, ..., A_m, b_m^\top]$, prefix $\in \mathbb{R}^{m(n+1) \times d}$. Then we apply our proposed reweighted method within several formal definitions, we first apply a linear regression on prefix with weight $W$ and bias $B$:

**Definition 4.2** (Weight and bias for reweight method). *We denote our weight $W \in \mathbb{R}^{m(n+1) \times m(n+1)}$ of reweight method that $W = \text{diag}(w)$ where $w \in \mathbb{R}^{m(n+1)}$ and additional bias $B \in \mathbb{R}^{m(n+1) \times d}$. We define $w^\top = \begin{bmatrix} (w_{a,1})^\top & (w_{b,1}) & \cdots & (w_{a,m})^\top & (w_{b,m}) \end{bmatrix}$, where $w_{a,i} \in \mathbb{R}^n$, $w_{b,i} \in \mathbb{R}$, then we define $B = \begin{bmatrix} (B_{a,1}) & (B_{b,1})^\top & \cdots & (B_{a,m}) & (B_{b,m}^\top) \end{bmatrix}$, where $B_{a,i} \in \mathbb{R}^{n \times d}$, $B_{b,i} \in \mathbb{R}^d$.*

We propose our definition of reweighted prefix

**Definition 4.3.** *Let $W \in \mathbb{R}^{m(n+1) \times m(n+1)}$ and $B \in \mathbb{R}^{m(n+1) \times d}$ be denoted as Definition 4.2, we define reweighted prefix as $\text{prefix}_{\text{reweight}} = W \cdot \text{prefix} + B$.*

We now provide a formal definition of the application of our method to the transformer architecture

**Definition 4.4** (Training objective of our reweight method in transformer). *Suppose that given embedded prompt prefix $:= [A_1, b_1, A_2, b_2, ..., A_m, b_m]$ and weight $W := \text{diag}(w)$, $w \in \mathbb{R}^{m(n+1)}$, bias $B \in \mathbb{R}^{m(n+1) \times d}$, given a clean and unbiased validation set $\mathcal{V} = \{(A_i^v, b_i^v)1 \le i \le |\mathcal{V}|\}$, where $|\mathcal{V}|$ represents the size of $\mathcal{V}$. We denote a language model with its parameters $\theta$ as $f_\theta(x)$, $f_\theta(x)$ implement in-context learning with prefix $W \cdot \text{prefix} + B$ as an in-context leaner, we denote it as $\text{ICL}_{\text{reweight}}(A)$. Given a training objective function $\mathcal{L}$ to train $f_\theta(x)$. We freeze $\theta$ and fine-tune $w$ to minimize*

$$\mathcal{L}_{valid} = \sum_{i=1}^{|\mathcal{V}|} \mathcal{L}(\text{ICL}_{\text{reweight}}(A_i^v), b_i^v)$$

To achieve Definition 4.2 and Definition 4.4, we propose Algorithm 1 (RICL), it can be seen as an algorithm for parameter-efficient-fine-tuning (PEFT) Hu et al. (2021); Lester et al. (2021); Li & Liang (2021); Liu et al. (2021b; 2022; 2023b); Wang et al. (2023b), where we freeze most of parameters of language model, to make language model implement unbiased in-context learning. Based on this, we have considered the propositions in Akyürek et al. (2022); Garg et al. (2022); Zhang et al. (2023b), we also proposed Agorithm 2 (LARICL), a fast approximation algorithm for optimal weight $w^*$ by utilizing closed-form solution of linear regression in Appendix C, we compare and discuss the effectiveness of two algorithms in experiments in Section 5.

---

**Algorithm 1** Reweight In-context learning (RICL)

---

    **Input:** Prompt $P$, validation set $\mathcal{V}$, learning rate $\alpha$, minimum error $\epsilon$
    **Output:** Optimal weights approximation $w^*$
1: **procedure** FINE-TUNETOREWEIGHTIN-CONTEXTLEARNING$(P, \mathcal{V}, \alpha, \epsilon)$
2:     We randomly initialize $w \in \mathbb{R}^{m(n+1)}$ from $\mathcal{N}(0, \mathbf{I}_{m(n+1)})$
3:     $W \leftarrow \mathrm{diag}(w)$
4:     $B \leftarrow O_{m(n+1) \times d}$
5:     $T \leftarrow O(1/\epsilon^2)$
6:     **for** $t = 1 \rightarrow T$ **do**
7:         $\mathcal{L}_{\mathrm{valid}} \leftarrow \sum_{i=1}^{|\mathcal{V}|} \mathcal{L}(\mathrm{ICL}_{\mathrm{reweight}}(A_i^v), b_i^v)$
8:         $W \leftarrow W - \alpha \nabla_W \mathcal{L}_{\mathrm{valid}}(W, B)$
9:         $B \leftarrow B - \alpha \nabla_B \mathcal{L}_{\mathrm{valid}}(W, B)$
10:    **end for**
11:    **return** $w$
12: **end procedure**

---

Then, we show our result that proving RICL is equivalent to Definition 3.5 with simple regularization.

**Lemma 4.5** (Regularization Version: equivalence between Definition 4.1 and Definition 4.4, informal version of Lemma A.1). *Let $\mathcal{L}_{\mathrm{valid}}^{(\mathrm{transformer})}$ and $\mathcal{L}_{\mathrm{valid}}^{(\mathrm{softmax})}$ be defined as Definition A.2, we have $W^*, B^* = \arg\min_{W,B} \mathcal{L}_{\mathrm{valid}}^{(\mathrm{transformer})}$ and $w^{(\mathrm{softmax})^*} = \arg\min_{w^{(\mathrm{softmax})}} \mathcal{L}_{\mathrm{valid}}^{(\mathrm{softmax})}$. Let $L_{\mathrm{reg}}$ be denoted as Definition A.4. If $\arg\min_{W,B}(\mathcal{L}^{(\mathrm{transformer})}(x) + L_{\mathrm{reg}}) \approx \arg\min_{W,B} L_{\mathrm{reg}}$, then for any $A \in \mathbb{R}^{n \times d}$ we have*

$$\mathrm{ICL}^{(\mathrm{transformer})}(A) = \mathrm{ICL}^{(\mathrm{softmax})}(A)$$

See Appendix A for the proof of Lemma 4.5.

### 4.3 CONVERGENCE OF THE REWEIGHTED TRAINING

Building upon the methodology established in Ren et al. (2018), we investigate the convergence of reweighted training. To substantiate our claims, we present empirical results that demonstrate two important properties of the reweighted training objective: 1) Lipschitz-smoothness of the gradient, and 2) a strict upper bound on the gradient. We first provide our result that the gradient of $L$ is Lipschitz-smooth:

**Lemma 4.6** (Lipschitz-smoothness of the gradient, Lemma 8.3 in Gao et al. (2023)). *We let $x, y \in \mathbb{R}^d$ with $\|x\|_2 \le R$ and $\|y\|_2 \le R$, where $R > 4$ denote a scalar. Let $x$ and $y$ satisfy $\max_{j \in [n]} \|A_{i[j],*}(x-y)\|_\infty < 0.01$, $\max_{j \in [n]} \|A_{i[j],*}\| \le R$, $\max_{j \in [n]} \|b_{i[j]}\|_2 \le 1$, we have Lipschitz-smooth property for $\nabla L$ as follows*

$$\|\nabla_x L(x, A_i, b_i) - \nabla_y L(y, A_i, b_i)\|_2 \le dn^2 \exp(5R^2) \cdot \|x - y\|_2$$

Then, we provide the result of the upper bound on the gradient below.

**Lemma 4.7** (Upper bound on gradient, informal version of Lemma B.9). *Given an input matrix $A_i \in \mathbb{R}^{n \times d}$ and a target vector $b \in \mathbb{R}^n$, where $A_i$ satisfies $\|A_i\| \le R$, $R > 4$, $b$ satisfies $\|b\|_2 \le 1$. Let $L(x, A_i, b_i)$ be defined as Definition 3.5, for all $x \in \mathbb{R}^d$, we have*

$$\|\nabla L(x, A_i, b_i)\|_2 \le 4R$$

Now that we have strictly proved conditions from the convergence guarantee, we present our result of the loss decrease in the reweighted training by running on Algorithm 1.

**Lemma 4.8** (Our version of Lemma 1 in Ren et al. (2018), inofrmal version of Lemma B.1). *Suppose the validation loss function is Lipschitz-smooth with constant $L$, where $L = dn^2 \exp(5R^2)$, and the train loss function of training data $A$ have $\sigma$-bounded gradients, where $\sigma = 4R$. Let the learning rate $\alpha_t$ satisfies $\alpha_t \leq \frac{2|\mathcal{B}|}{L\sigma^2}$, where $|\mathcal{B}|$ is the training batch size of batch $\mathcal{B}$ of validation set $\mathcal{V}$. Then, following our algorithm, the validation loss always monotonically decreases for any sequence of training batches, namely,*

$$\mathcal{L}_{\text{valid}}(w_{t+1}) \leq \mathcal{L}_{\text{valid}}(w_t)$$

*where $\mathcal{L}_{\text{valid}}(w)$ is the total validation loss in Definition 4.1.*

*Furthermore, in expectation, the $\mathcal{L}_{\text{valid}}(w)$ holds only when the gradient of validation loss becomes $0$ at some time step $t$, namely $\mathbb{E}_t[\mathcal{L}_{\text{valid}}(w_{t+1})] = \mathcal{L}_{\text{valid}}(w)$ if and only if $\nabla \mathcal{L}_{\text{valid}}(w) = 0$, where the expectation is taking over possible training batches at time step $t$.*

Moreover, we can prove the convergence rate of our method to be $O(1/\epsilon^2)$.

**Theorem 4.9** (Our version of Theorem 2 in Ren et al. (2018), informal version of Theorem B.2). *Suppose $\mathcal{L}_{\text{unbiased}}$, $L(x, A_i, b_i)$ and $\alpha_t$ satisfy the aforementioned conditions, then the Algorithm 1 achieves $\mathbb{E}[\|\nabla \mathcal{L}_{\text{valid}}(w)\|^2] \leq \epsilon$ in $O(1/\epsilon^2)$ steps. More specifically,*

$$\min_{0 \leq t \leq T} \mathbb{E}[\|\mathcal{L}_{\text{valid}}(w_t)\|^2] \leq \frac{C}{\sqrt{T}}$$

*where $C$ is some constant independent of the convergence process.*

Proofs of Lemma 4.6, Lemma 4.7, Lemma 4.8 and Theorem 4.9 can be saw in Appendix B.

## 5 RESULTS

In this section, we present our experimental results, proving that our reweight algorithms can approximate unbiased in-context learning. We first pretrained a GPT-2 to learn in-context, then trained it on prefixes with different distributions and compared the performances of our algorithm between vanilla ICL. For the details about experiment setup, please refer to Appendix D.

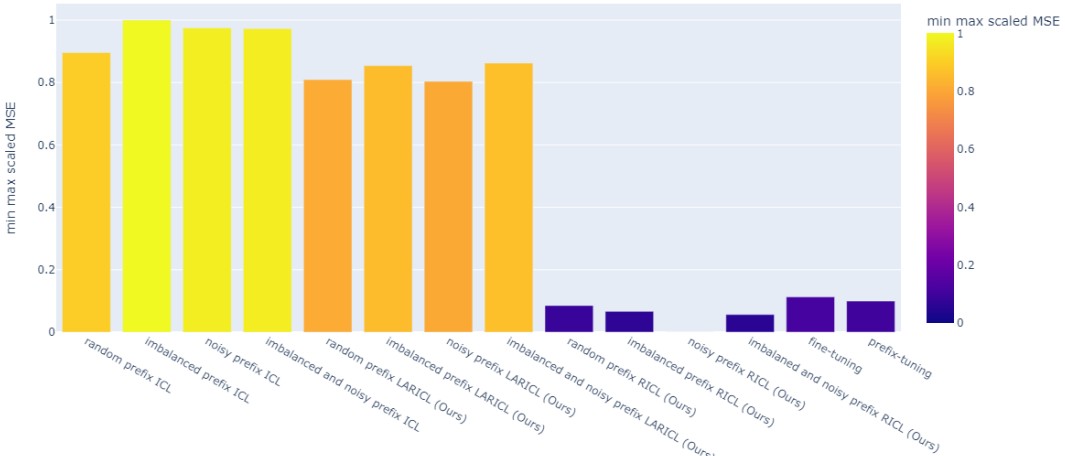

Figure 2: Comparison of performances (min-max-scaled-mean-squared error (MSE), the smaller the better) of our methods (RICL, LARICL) with performances of ICL method, fine-tuning method, and prefix-tuning method on different distribution prefixes.

**The performances of reweight algorithms on prefixes with different distributions.** Figure 2 shows the performances of Algorithm 1 and Algorithm 2 on random prefix (randomly select input-output pair from unbiased and clean dataset), imbalanced prefix (randomly select input-output pair

from imbalanced distribution), noisy prefix (add random noise on the output of each example), imbalanced and noisy prefix (combine imbalanced prefix and noisy prefix). We use min-max-scaled mean-squared error (MSE) (See Metrics part of Appendix D.1 for a detailed explanation of min-max-scaled MSE) to measure the performance of our methods, where the minimum value is 0 and the maximum value is 1 for comparison. We fine-tune the pre-trained GPT-2 on the validation set and record its performance on the test set, we can conclude from Figure 2 that our method (RICL) has a performance that is better than the performance of the fine-tuning method and prefix-tuning method Li & Liang (2021) on numerical datasets. At the same time, RICL has fewer parameter challenges and better generalization since the reweighted training.

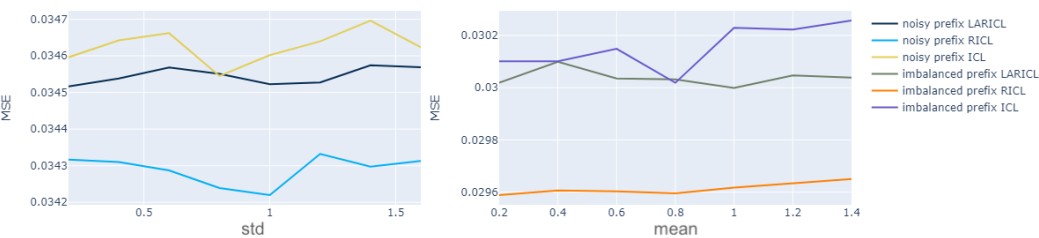

Figure 3: Performance (Mean-squared error (MSE), the smaller the better) of our methods on imbalanced prefix and noisy prefix.

**Robustness on imbalanced prefix and noisy prefix.**    To introduce noise into the prefixes, we randomly sampled noise values from $\varepsilon \in \mathcal{N}(0, \mathrm{std} \cdot \mathbf{I}_n)$. We then conducted experiments on different prefixes, varying the standard deviation parameter, denoted as "std" in the left image of Figure 3. The results clearly indicate that as the noise in the prompt increases, the performance of ICL is significantly affected. However, our proposed methods, RICL and LARICL, maintain better performance consistently, despite the increasing noise levels. This demonstrates the robustness of our methods in handling noisy prefixes.

Similarly, to evaluate the performance under imbalanced conditions, we generated imbalanced prompts by sampling from a normal distribution with a specified mean value, denoted as $\mathcal{N}(\mathrm{mean}, \mathbf{I}_d)$. We conducted experiments on different prefixes, varying the mean parameter, denoted as "mean" in the right image of Figure 3. Notably, our methods consistently maintained stable performances across the varying mean values, showcasing their ability to handle imbalanced data effectively.

These findings emphasize the superiority of our proposed approaches, RICL and LARICL, in maintaining robust and stable performance even in the presence of noisy and imbalanced prefixes. By addressing these challenges, our methods provide a reliable solution for enhancing the performance of in-context learning models in real-world scenarios.

## 6 CONCLUSION

Our research focuses on addressing the challenge of imbalanced in-context learning in language models. We propose a novel fine-tuning algorithm that effectively tackles this issue by incorporating reweighting of input prompts. We highlight the significance of understanding softmax regression in language model inference for successful in-context learning. Moreover, we rigorously prove the convergence of our method and establish its stability properties, demonstrating that gradient descent methods reliably converge to the optimal solution.

Extensive experimentation validates the effectiveness of our proposed Reweighted In-Context Learning (RICL) approach, surpassing the performance of other existing methods such as fine-tuning, prefix-tuning, and casual in-context learning. To demonstrate the robustness of our algorithm, we conduct comprehensive tests using varying degrees of imbalanced data and noisy prefixes. The results of our experiments clearly showcase the significant improvements achieved by RICL in addressing imbalanced in-context learning challenges. By outperforming other methods, RICL demonstrates its efficacy in handling different levels of data imbalance and noisy input prompts.

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
