## A    PROOF OF EQUIVALENCE BETWEEN DEFINITION 4.1 AND DEFINITION 4.4

### A.1    REGULARIZATION VERSION: EQUIVALENCE BETWEEN DEFINITION 4.1 AND DEFINITION 4.4

**Lemma A.1** (Regularization Version: equivalence between Definition 4.1 and Definition 4.4, formal version of Lemma 4.5). *Let* $\mathcal{L}_{\text{valid}}^{(\text{transformer})}$ *and* $\mathcal{L}_{\text{valid}}^{(\text{softmax})}$ *be defined as Definition A.2, we have* $W^*, B^* = \arg\min_{W,B} \mathcal{L}_{\text{valid}}^{(\text{transformer})}$ *and* $w^{(\text{softmax})^*} = \arg\min_{w^{(\text{softmax})}} \mathcal{L}_{\text{valid}}^{(\text{softmax})}$. *Let* $L_{\text{reg}}$ *be denoted as Definition A.4. For any* $A \in \mathbb{R}^{n \times d}$ *we have*

$$\text{ICL}^{(transformer)}(A) = \text{ICL}^{(softmax)}(A)$$

*with assumptions*

- $\arg\min_{W,B}(\mathcal{L}^{(\text{transformer})}(x) + L_{\text{reg}}) \approx \arg\min_{W,B} L_{\text{reg}}$

*Proof.* Following $\arg\min_{W,B}(\mathcal{L}^{(\text{transformer})}(x) + L_{\text{reg}}) \approx \arg\min_{W,B} L_{\text{reg}}$, we have

$$\text{diag}(w_{a,i}^{(\text{transformer})})A + B_{a,i}^{(\text{transformer})} = A_i$$

and

$$B_{b,i}^{(\text{transformer})} = (\sqrt{w_{b,i}^{(\text{transformer})}} - 1) \cdot \langle \exp(A_i^\top x), \mathbf{1}_n \rangle^{-1} \exp(A_i^\top x)$$

for all $w_{a,i}^{(\text{transformer})}, B_{a,i}^{(\text{transformer})}$ and $w_{b,i}^{(\text{transformer})}, B_{b,i}^{(\text{transformer})}$ in $W, B$.

When $w_i^{(\text{softmax})} = w_{b,i}^{(\text{transformer})}$, we have

$$\mathcal{L}^{(\text{softmax})}(x) = \mathcal{L}^{(\text{transformer})}(x)$$

where this equality uses Lemma A.5.

So we have

$$x^{(\text{transformer})} = x^{(\text{softmax})}$$

where this equality uses the definitions of $x^{(\text{transformer})}$ and $x^{(\text{softmax})}$.

Then,

$$\text{ICL}^{(transformer)}(A) = \text{ICL}^{(softmax)}(A)$$

where this equality uses the definitions of $\text{ICL}^{(transformer)}(A)$ and $\text{ICL}^{(softmax)}(A)$.    $\square$

**Definition A.2** (Validation loss). *Let* $\text{ICL}^{(transformer)}(A)$ *and* $\text{ICL}^{(softmax)}(A)$ *be defined as Definition A.3. Suppose that given a clean and unbiased validation set* $\mathcal{V} = \{(A_i^v, b_i^v)1 \le i \le |\mathcal{V}|\}$, *where* $|\mathcal{V}|$ *represents the size of* $\mathcal{V}$. *We define*

$$\mathcal{L}_{\text{valid}}^{(\text{transformer})} := \sum_{i=1}^{|\mathcal{V}|} L(x^{(\text{transformer})}, A_i^v, b_i^v)$$

*and we define*

$$\mathcal{L}_{\text{valid}}^{(\text{softmax})} := \sum_{i=1}^{|\mathcal{V}|} L(x^{(\text{softmax})}, A_i^v, b_i^v)$$

**Definition A.3** (Definitions of $\text{ICL}^{(transformer)}(A)$ and $\text{ICL}^{(softmax)}(A)$). *Let* $\mathcal{L}^{(\text{softmax})}(x)$ *and* $\mathcal{L}^{(\text{transformer})}(x)$ *be defined as Definition A.5. Let* $f(x)$ *be defined as Definition 3.1. We define* $L_{\text{reg}}$ *as Definition A.4. Following Definition A.4, we define*

$$\mathcal{L}_{\text{reg}} := \gamma \cdot \sum_{i=1}^n (\| \text{diag}(w_{a,i}^{(\text{transformer})})A_i + B_{a,i}^{(\text{transformer})} - A_i \|^2$$

$$+ \|B_{b,i}^{(\text{transformer})} - (\sqrt{w_{b,i}^{(\text{transformer})}} - 1) \cdot b_i\|_2^2)$$

*Since Theorem 3.4, we define*

$$\text{ICL}^{(transformer)}(A) = f(x^{(\text{transformer})})$$

*where $x^{(\text{transformer})} := \arg\min_x (\mathcal{L}^{(\text{transformer})}(x) + \mathcal{L}_{\text{reg}})$. And we define*

$$\text{ICL}^{(softmax)}(A) = f(x^{(\text{softmax})})$$

*where $x^{(\text{softmax})} := \arg\min_x \mathcal{L}^{(\text{softmax})}(x)$.*

**Definition A.4.** *We define a regularization term for approximate the conditions in Lemma A.5. We define*

$$L_{\text{reg}} := \gamma(\|\operatorname{diag}(w_a^{(\text{transformer})})A + B_a^{(\text{transformer})} - A\|^2$$

$$+ \|B_b^{(\text{transformer})} - (\sqrt{w_b^{(\text{transformer})}} - 1) \cdot b\|_2^2)$$

*where $\gamma > 0$ be denote a constant.*

## A.2 Conditional Constraint Version: Equivalence between Definition 4.1 and Definition 4.4 in Multiple Input-output Pairs

**Lemma A.5** (Conditional constraint version: equivalence between Definition 4.1 and Definition 4.4 in multiple input-output pairs)**.** *If the following conditions hold*

- *Let $\mathcal{L}^{(\text{softmax})}(x)$ and $\mathcal{L}^{(\text{transformer})}(x)$ be defined as Definition A.6*

- $\operatorname{diag}(w_a^{(\text{transformer})})A + B_a^{(\text{transformer})} = A$

- $w_b^{(\text{transformer})} = \sqrt{w^{(\text{softmax})}}$

- $B_b^{(\text{transformer})} = (\sqrt{w^{(\text{softmax})}} - 1) \cdot b \approx (\sqrt{w^{(\text{softmax})}} - 1) \cdot \langle \exp(A^\top x), \mathbf{1}_n\rangle^{-1} \exp(A^\top x)$

*We have*

$$\mathcal{L}^{(\text{softmax})}(x) = \mathcal{L}^{(\text{transformer})}(x)$$

*Proof.* We have

$$\mathcal{L}^{(\text{softmax})}(x) = \sum_{i=1}^{m} w_i^{(\text{softmax})} \|\langle \exp(A_i^\top x), \mathbf{1}_n\rangle^{-1} \exp(A_i^\top x) - b_i\|_2^2$$

$$= \sum_{i=1}^{m} \|\langle \exp(A_{i_{\text{reweight}}}^\top x), \mathbf{1}_n\rangle^{-1} \exp(A_{i_{\text{reweight}}}^\top x) - b_{i_{\text{reweight}}}\|_2^2$$

$$= \mathcal{L}^{(\text{transformer})}(x)$$

where the first equality uses the definition of $\mathcal{L}^{(\text{softmax})}(x)$, the second equality uses Lemma A.7, the third equality uses the definition of $\mathcal{L}^{(\text{transformer})}(x)$. $\square$

**Definition A.6** (In-context learning with $m$ input-output pairs)**.** *We discuss the case of multiple input-output pairs. Following Lemma A.7, we define*

$$\mathcal{L}^{(\text{softmax})}(x) := \sum_{i=1}^{m} w_i^{(\text{softmax})} \|\langle \exp(A_i^\top x), \mathbf{1}_n\rangle^{-1} \exp(A_i^\top x) - b_i\|_2^2$$

*and we define*

$$\mathcal{L}^{(\text{transformer})}(x) = \sum_{i=1}^{m} \|\langle \exp(A_{i_{\text{reweight}}}^\top x), \mathbf{1}_n\rangle^{-1} \exp(A_{i_{\text{reweight}}}^\top x) - b_{i_{\text{reweight}}}\|_2^2$$

*where $A_{i_{\text{reweight}}} = \operatorname{diag}(w_{a,i}^{(\text{transformer})})A_i + B_{a,i}^{(\text{transformer})}$ and $b_{i_{\text{reweight}}} = w_{b,i}^{(\text{transformer})}b + B_{b,i}^{(\text{transformer})}$.*

## A.3 CONDITIONAL CONSTRAINT VERSION: EQUIVALENCE BETWEEN DEFINITION 4.1 AND DEFINITION 4.4 IN SINGLE INPUT-OUTPUT PAIR

**Lemma A.7** (Conditional constraint version: equivalence between Definition 4.1 and Definition 4.4 in single input-output pair). *We discuss the case of a single input-output pair, let loss of single softmax regression in Definition 4.1 be defined as $L^{(\mathrm{softmax})}(x) = w^{(\mathrm{softmax})}\|\langle \exp(A^\top x), \mathbf{1}_n\rangle^{-1}\exp(A^\top x) - b\|_2^2$. Let loss of single pair in in-context learning of transformer embedding layer in Definition 4.4 be defined as $L^{(\mathrm{transformer})}(x) = \|\langle\exp(A_{\mathrm{rewight}}^\top x), \mathbf{1}_n\rangle^{-1}\exp(A_{\mathrm{rewight}}^\top x) - b_{\mathrm{rewight}}\|_2^2$, where $A_{\mathrm{rewight}} = \mathrm{diag}(w_a^{(\mathrm{transformer})})A + B_a^{(\mathrm{transformer})}$ and $b_{\mathrm{rewight}} = w_b^{(\mathrm{transformer})}b + B_b^{(\mathrm{transformer})}$. We have*

$$L^{(\mathrm{softmax})}(x) = L^{(\mathrm{transformer})}(x)$$

*when satisfies the following conditions*

- $\mathrm{diag}(w_a^{(\mathrm{transformer})})A + B_a^{(\mathrm{transformer})} = A$

- $w_b^{(\mathrm{transformer})} = \sqrt{w^{(\mathrm{softmax})}}$

- $B_b^{(\mathrm{transformer})} = (\sqrt{w^{(\mathrm{softmax})}}-1)\cdot b \approx (\sqrt{w^{(\mathrm{softmax})}}-1)\cdot\langle\exp(A^\top x), \mathbf{1}_n\rangle^{-1}\exp(A^\top x)$

*Proof.* We assume that

$$\mathrm{diag}(w_a^{(\mathrm{transformer})})A + B_a^{(\mathrm{transformer})} = A \tag{2}$$

and $w_b^{(\mathrm{transformer})}$ satisfies that

$$w_b^{(\mathrm{transformer})} = \sqrt{w^{(\mathrm{softmax})}} \tag{3}$$

and $B_a^{(\mathrm{transformer})}$ satisfies that

$$B_b^{(\mathrm{transformer})} = (\sqrt{w^{(\mathrm{softmax})}} - 1)\cdot\langle\exp(A^\top x), \mathbf{1}_n\rangle^{-1}\exp(A^\top x) \tag{4}$$

As we satisfy the tree conditions above, we have

$$
\begin{aligned}
L^{(\mathrm{transformer})}(x) &= \|\langle\exp(A_{\mathrm{rewight}}^\top x), \mathbf{1}_n\rangle^{-1}\exp(A_{\mathrm{rewight}}^\top x) - b_{\mathrm{rewight}}\|_2^2 \\
&= \|\langle\exp((\mathrm{diag}(w_a^{(\mathrm{transformer})})A + B_a^{(\mathrm{transformer})})^\top x), \mathbf{1}_n\rangle^{-1} \\
&\quad \cdot \exp((\mathrm{diag}(w_a^{(\mathrm{transformer})})A + B_a^{(\mathrm{transformer})})^\top x) - b_{\mathrm{rewight}}\|_2^2 \\
&= \|\langle\exp(A^\top x), \mathbf{1}_n\rangle^{-1}\exp(A^\top x) - b_{\mathrm{rewight}}\|_2^2 \\
&= \|\langle\exp(A^\top x), \mathbf{1}_n\rangle^{-1}\exp(A^\top x) - w_b^{(\mathrm{transformer})}b + B_b^{(\mathrm{transformer})}\|_2^2 \\
&= \|\langle\exp(A^\top x), \mathbf{1}_n\rangle^{-1}\exp(A^\top x) - \sqrt{w^{(\mathrm{softmax})}}b + B_b^{(\mathrm{transformer})}\|_2^2 \\
&= \|\langle\exp(A^\top x), \mathbf{1}_n\rangle^{-1}\exp(A^\top x) - \sqrt{w^{(\mathrm{softmax})}}b \\
&\quad + (\sqrt{w^{(\mathrm{softmax})}} - 1)\cdot\langle\exp(A^\top x), \mathbf{1}_n\rangle^{-1}\exp(A^\top x)\|_2^2 \\
&= \|\sqrt{w^{(\mathrm{softmax})}}\cdot\langle\exp(A^\top x), \mathbf{1}_n\rangle^{-1}\exp(A^\top x) - \sqrt{w^{(\mathrm{softmax})}}b\|_2^2 \\
&= w^{(\mathrm{softmax})}\cdot\|\langle\exp(A^\top x), \mathbf{1}_n\rangle^{-1}\exp(A^\top x) - b\|_2^2 \\
&= L^{(\mathrm{softmax})}(x)
\end{aligned}
$$

where the first equality uses the definition of $L^{(\mathrm{transformer})}(x)$, the second equality uses the definition of $A_{\mathrm{rewight}}$, the third equality uses Eq. (2), the fourth equality uses the definition of $b_{\mathrm{rewight}}$, the fifth equality uses Eq. (3), the sixth equality uses Eq. (4), the seventh, eighth equalities uses simple algebras, the ninth equality uses the definition of $L^{(\mathrm{softmax})}(x)$. □

### A.4 CALCULATION RESULTS OF EACH INPUT-OUTPUT PAIR IN Prefix$_{\text{reweight}}$

**Lemma A.8** (Calculation results of each input-output pair in prefix$_{\text{reweight}}$). *Let $W \in \mathbb{R}^{m(n+1) \times m(n+1)}$ and $B \in \mathbb{R}^{m(n+1) \times d}$ be denoted as Definition 4.2, given prefix $:= [A_1, b_1^\top, A_2, b_2^\top, ..., A_m, b_m^\top]$, prefix $\in \mathbb{R}^{m(n+1) \times d}$, let prefix$_{\text{reweight}}$ be defined as Definition 4.3, we have*

$$\text{prefix}_{\text{reweight}} = [\text{diag}(w_{a,1})A_1 + B_{a,1}, w_{b,1}b_1 + B_{b,1}, ..., \text{diag}(w_{a,m})A_m + B_{a,m}, w_{b,m}b_m + B_{a,m}]$$

*Proof.* We have

$$w^\top = \begin{bmatrix} (w_{a,1})^\top & (w_{b,1}) & \cdots & (w_{a,m})^\top & (w_{b,m}) \end{bmatrix} \tag{5}$$

where $w_{a,i} \in \mathbb{R}^n$, $w_{b,i} \in \mathbb{R}$.

Also, we have

$$B = \begin{bmatrix} (B_{a,1}) & (B_{b,1})^\top & \cdots & (B_{a,m}) & (B_{b,m})^\top \end{bmatrix} \tag{6}$$

where $B_{a,i} \in \mathbb{R}^{n \times d}$, $B_{b,i} \in \mathbb{R}^d$.

We can show that

$$\begin{aligned}
&\text{prefix}_{\text{reweight}} \\
&= W \cdot \text{prefix} + B \\
&= \text{diag}(w) \cdot \text{prefix} + B \\
&= [\text{diag}(w_{a,1})A_1 + B_{a,1}, w_{b,1}b_1 + B_{b,1}, ..., \text{diag}(w_{a,m})A_m + B_{a,m}, w_{b,m}b_m + B_{a,m}]
\end{aligned}$$

where the first equality uses Definition 4.3, the second equality uses the definition of $W$, the third equality uses simple algebra and Eq. (5) and Eq. (6). $\square$

## B PROOF OF CONVERGENCE OF REWEIGHTED TRAINING

### B.1 MAIN RESULT

**Lemma B.1** (Our version of Lemma B.5, formal version of Lemma 4.8). *Suppose the validation loss function is Lipschitz-smooth with constant $L$, where $L = dn^2 \exp(5R^2)$, and the train loss function of training data $A$ have $\sigma$-bounded gradients, where $\sigma = 4R$. Let the learning rate $\alpha_t$ satisfies $\alpha_t \leq \frac{2|\mathcal{B}|}{L\sigma^2}$, where $|\mathcal{B}|$ is the training batch size of batch $\mathcal{B}$ of validation set $\mathcal{V}$. Then, following our algorithm, the validation loss always monotonically decreases for any sequence of training batches, namely,*

$$\mathcal{L}_{valid}(w_{t+1}) \leq \mathcal{L}_{valid}(w_t)$$

*where $\mathcal{L}_{valid}(w)$ is the total validation loss in Definition 4.1.*

*Furthermore, in expectation, the $\mathcal{L}_{\text{valid}}(w)$ holds only when the gradient of validation loss becomes $0$ at some time step $t$, namely $\mathbb{E}_t[\mathcal{L}_{\text{valid}}(w_{t+1})] = \mathcal{L}_{\text{valid}}(w)$ if and only if $\nabla \mathcal{L}_{\text{valid}}(w) = 0$, where the expectation is taking over possible training batches at time step $t$.*

*Proof.* Since $L$ and $\sigma$ have been strictly bounded in Lemma B.8 and Lemma B.9, this proof follows from Lemma B.5. $\square$

**Theorem B.2** (Our version of Theorem B.6, formal version of Therorem 4.9). *Suppose $\mathcal{L}_{unbiased}$, $L(x, A_i, b_i)$ and $\alpha_t$ satisfy the aforementioned conditions, then the Algorithm 1 achieves $\mathbb{E}[\|\nabla \mathcal{L}_{valid}(w)\|^2] \leq \epsilon$ in $O(1/\epsilon^2)$ steps. More specifically,*

$$\min_{0 \leq t \leq T} \mathbb{E}[\|\mathcal{L}_{valid}(w_t)\|^2] \leq \frac{C}{\sqrt{T}}$$

*where $C$ is some constant independent of the convergence process.*

*Proof.* This proof follows from Theorem B.6 and Lemma B.1. $\square$

### B.2 REWEIGHTED TRAINING ANALYSIS

**Definition B.3** (Convergence condition in Ren et al. (2018)). *A function $f(x) : \mathbb{R}^d \to \mathbb{R}$ is said to be Lipschitz-smooth with constant $L$ if*

$$\|\nabla f(x) - \nabla f(y)\| \le L\|x - y\|, \forall x, y \in \mathbb{R}^d$$

**Definition B.4** (Convergence condition in Ren et al. (2018)). *$f(x)$ has $\sigma$-bounded gradients if $\|\nabla f(x)\| \le \sigma$ for all $x \in \mathbb{R}^d$.*

**Lemma B.5** (Lemma 1 in Ren et al. (2018)). *Suppose the validation loss function is Lipschitz-smooth with constant $L$, and the train loss function of training data $x_i$ have $\sigma$-bounded gradients. Let the learning rate $\alpha_t$ satisfies $\alpha_t \le \frac{2n}{L\sigma^2}$, where $n$ is the training batch size. Then, following our algorithm, the validation loss always monotonically decreases for any sequence of training batches, namely,*

$$G(\theta_{t+1}) \le G(\theta)$$

*where $G(\theta)$ is the total validation loss*

$$G(\theta) = \frac{1}{M} \sum_{i=1}^M f_i^v(\theta_{t+1}(\epsilon))$$

*Furthermore, in expectation, the $\mathcal{L}_{\text{valid}}(w)$ holds only when the gradient of validation loss becomes $0$ at some time step $t$, namely $\mathbb{E}_t[\mathcal{L}_{\text{valid}}(w_{t+1})] = \mathcal{L}_{\text{valid}}(w)$ if and only if $\nabla \mathcal{L}_{\text{valid}}(w) = 0$, where the expectation is taking over possible training batches at time step $t$.*

**Theorem B.6** (Theorem 2 in Ren et al. (2018)). *Suppose $G$, $f_i$ and $\alpha_t$ satisfy the aforementioned conditions, then the Algorithm 1 achieves $\mathbb{E}[\|\nabla G(\theta_t)\|^2] \le \epsilon$ in $O(1/\epsilon^2)$ steps. More specifically,*

$$\min_{0 < t < T} \mathbb{E}[\|\nabla G(\theta_t)\|^2] \le \frac{C}{\sqrt{T}}$$

*where $C$ is some constant independent of the convergence process.*

### B.3 GRADIENT OF $L(x, A_i, b_i)$

**Lemma B.7** (Deng et al. (2023); Gao et al. (2023); Chu et al. (2023)). *Let $L(x, A_i, b_i)$ be defined as Definition 3.5, we have*

$$\nabla L(x, A_i, b_i) = A_{i*,j}^\top (-f_i(x)(f_i(x) - b_i)^\top f_i(x) + \text{diag}(f_i(x))(f_i(x) - b_i))$$

*where $j \in [n]$ denote a integer.*

### B.4 LIPSCHITZ PROPERTY FOR $\nabla L$

**Lemma B.8** (Lemma 8.3 in Gao et al. (2023)). *We let $x, y \in \mathbb{R}^d$ with $\|x\|_2 \le R$ and $\|y\|_2 \le R$, where $R > 4$ denote a scalar. Let $x$ and $y$ satisfy $\max_{j \in [n]} \|A_{i[j],*}(x-y)\|_\infty < 0.01$, $\max_{j \in [n]} \|A_{i[j],*}\| \le R$, $\max_{j \in [n]} \|b_{i[j]}\|_2 \le 1$, we have Lipschitz-smooth property for $\nabla L$ as follows*

$$\|\nabla_x L(x, A_i, b_i) - \nabla_y L(y, A_i, b_i)\|_2 \le dn^2 \exp(5R^2) \cdot \|x - y\|_2$$

### B.5 UPPER BOUND ON $\nabla L(x, A_i, b_i)$

**Lemma B.9** (Formal version of Lemma 4.7). *Given an input matrix $A_i \in \mathbb{R}^{n \times d}$ and a target vector $b \in \mathbb{R}^n$, where $A_i$ satisfies $\|A_i\| \le R$, $R > 4$, $b$ satisfies $\|b\|_2 \le 1$. Let $L(x, A_i, b_i)$ be defined as Definition 3.5, for all $x \in \mathbb{R}^d$, we have*

$$\|\nabla L(x, A_i, b_i)\|_2 \le 4R$$

*Proof.* We have

$$\|\nabla L(x, A_i, b_i)\|_2 \le \|A_{i*,j}^\top (-f_i(x)(f_i(x) - b_i)^\top f_i(x) + \text{diag}(f_i(x))(f_i(x) - b_i))\|_2$$
$$\le \|A\| \cdot (\| - f_i(x)(f_i(x) - b_i)^\top f_i(x) + \text{diag}(f_i(x))(f_i(x) - b_i))\|_2)$$

$$\leq \|A\| \cdot (\|f_i(x)(f_i(x) - b_i)^\top f_i(x)\|_2 + \|\operatorname{diag}(f_i(x))(f_i(x) - b_i)\|_2)$$
$$\leq \|A\| \cdot (\|f_i(x)\|_2 \cdot |(f_i(x) - b_i)^\top f_i(x)| + \|\operatorname{diag}(f_i(x))(f_i(x) - b_i)\|_2)$$
$$\leq \|A\| \cdot (\|f_i(x)\|_2 \cdot \|f_i(x) - b_i\|_2 \cdot \|f_i(x)\|_2 + \|\operatorname{diag}(f_i(x))(f_i(x) - b_i)\|_2)$$
$$\leq \|A\| \cdot (\|f_i(x)\|_2 \cdot \|f_i(x) - b_i\|_2 \cdot \|f_i(x)\|_2 + \|\operatorname{diag}(f_i(x))\| \cdot \|f_i(x) - b_i\|_2)$$
$$\leq \|A\| \cdot (\|f_i(x)\|_2 \cdot \|f_i(x) - b_i\|_2 \cdot \|f_i(x)\|_2 + \|f_i(x)\|_2 \cdot \|f_i(x) - b_i\|_2)$$
$$\leq \|A\| \cdot (\|f_i(x) - b_i\|_2 + \|f_i(x) - b_i\|_2)$$
$$\leq 4\|A\|$$
$$\leq 4R$$

where the first equality uses Lemma B.7, the second equality uses simple algebra, the third, fourth, fifth, sixth, seventh equalities use Fact B.10, the eighth equality uses $\|f(x)\|_2 \leq 1$, the ninth equality uses Lemma B.11, the tenth equality uses $\|A\| \leq R$. □

### B.6 BASIC BOUNDS

**Fact B.10.** *Denote $u, v \in \mathbb{R}^n$ denote two vectors such that*

- $|u^\top v| \leq \|u\|_2 \cdot \|v\|_2$

- $\|\operatorname{diag}(u)\| \leq \|u\|_2$

- $\|u + v\|_2 \leq \|u\|_2 + \|v\|_2$

- *Denote $\alpha \in \mathbb{R}$ a scalar, $\|\alpha u\|_2 = |\alpha| \cdot \|u\|_2$*

**Lemma B.11** (Part 1 of Lemma 66 in Chu et al. (2023)). *Let $f(x)$ be defined as Definition 3.1, $\|b\|_2 \leq 1$, we have*

$$\|f(x) - b\|_2 \leq 2$$

## C A FAST WEIGHT APPROXIMATION ALGORITHM UNDER LINEAR REGRESSION LOSS

Here we follow the results in Akyürek et al. (2022); Garg et al. (2022); Zhang et al. (2023b); Von Oswald et al. (2023), under the assumption that transformer learns linear regression in in-context learning, we provide our definition for linear in-context learning below.

**Definition C.1** (In-context learning loss under linear regression loss). *Suppose there is an input prompt $P = \{(A_1, b_1), (A_2, b_2), ..., (A_m, b_m)\}$ with $m$ data points $(A_i, b_i) \in \mathbb{R}^{n \times d} \times \mathbb{R}^n$ for all $i \in [m]$. We define in-context learning loss as follows*

$$\mathcal{L} := \sum_{i=1}^{m} L(x, A_i, b_i)$$

*where $L(x, A_i, b_i)$ is single in-context learning loss such that $L(x, A_i, b_i) := 0.5\|A_i x - b_i\|_2^2$.*

Then we have closed-form solution for Definition C.1 that $x = (A^\top A)^{-1} A^\top b$. We show our definition of reweight method on linear in-context learning as follows:

**Definition C.2** (Weight for reweight method). *We denote our weight $W \in \mathbb{R}^{m(n+1) \times m(n+1)}$ of reweight method that $W = \operatorname{diag}(w)$ where $w \in \mathbb{R}^{m(n+1)}$ and additional bias $B \in \mathbb{R}^{m(n+1) \times d}$. We define $w^\top = [(w_{a,1})^\top \quad (w_{b,1}) \quad \cdots \quad (w_{a,m})^\top \quad (w_{b,m})]$, where $w_{a,i} \in \mathbb{R}^n$, $w_{b,i} \in \mathbb{R}$.*

And we propose our definition of reweighted preifx

**Definition C.3.** *Let $W \in \mathbb{R}^{m(n+1) \times m(n+1)}$ be denoted as Definition C.2, we define reweighted prefix as $\operatorname{prefix}_{\text{reweight}} = W \cdot \operatorname{prefix}$.*

We now provide a formal definition of the application of our method to the transformer architecture

**Definition C.4** (Training objective of our reweight method on linear in-context learning). *Suppose that given embedded prompt* prefix := $[A_1, b_1, A_2, b_2, ..., A_m, b_m]$ *and weight* $w \in \mathbb{R}^{m(n+1)}$, *given a clean and unbiased validation set* $\mathcal{V} = \{(A_i^v, b_i^v) 1 \leq i \leq |\mathcal{V}|\}$, *where* $|\mathcal{V}|$ *represents the size of* $\mathcal{V}$. *We denote a language model with its parameters* $\theta$ *as* $f_\theta(x)$, $f_\theta(x)$ *implement in-context learning with prefix* $W \cdot$ prefix $+ B$ *as an in-context leaner, we denote it as* $\mathrm{ICL}_{\mathrm{reweight}}(A)$. *Given a training objective function* $\mathcal{L}$ *to train* $f_\theta(x)$. *We freeze* $\theta$ *and fine-tune* $w$ *to minimize*

$$\mathcal{L}_{\mathrm{valid}} = \sum_{i=1}^{|\mathcal{V}|} \mathcal{L}(\mathrm{ICL}_{\mathrm{reweight}}(A_i^v), b_i^v)$$

*where* $\mathrm{ICL}_{\mathrm{reweight}}(A_i^v) = A_i^v(A^\top A)^{-1}A^\top b$, $A = \frac{1}{m}\sum_{i=1}^m \mathrm{diag}(w_{a,i})A_i$, $b = \frac{1}{m}\sum_{i=1}^m w_{b,i}b_i$.

Hence, we can train weight $w$ by

---

**Algorithm 2** Linear approximation reweighted in-context learning (LARICL)

---

    **Input:** Prompt $P$, validation set $\mathcal{V}$, learning rate $\alpha$, minimum error $\epsilon$
    **Output:** Optimal weights approximation $w^*$
1: **procedure** OPTIMALWEIGHTSLINEARAPPROXIMATION($P, \mathcal{V}, \alpha, \epsilon$)
2:     $\epsilon \leftarrow \mathbf{1}_{m(n+1)}$
3:     $T \leftarrow O(1/\epsilon^2)$
4:     **for** $t = 1 \rightarrow T$ **do**
5:         $A \leftarrow \frac{1}{m}\sum_{i=1}^m \mathrm{diag}(w_{a,i})A_i$
6:         $b \leftarrow \frac{1}{m}\sum_{i=1}^m w_{b,i}b_i$
7:         $\mathcal{L}_{valid} \leftarrow \sum_{i=1}^{|\mathcal{V}|}\|A_i^v(A^\top A)^{-1}A^\top b - b_i^v\|_2^2$
8:         $w \leftarrow w - \alpha\nabla_w\mathcal{L}_{valid}(w)$
9:     **end for**
10:    **return** $w$
11: **end procedure**

---

## D   EXPERIMENTAL DETAILS

### D.1   SETUP

**Pretrain GPT-2 to Learn In-context.**   We pre-train a GPT-2 with 12 layers, 8 heads, dimension $d = 128$ and max token length n_positions $= 1024$ under softmax regreesion (Definition 3.1). We let $n = 16$ for each input-output pairs, we made 600000 data for pre-training and each data has 40 shots of input-output pairs, where the input $A_i \in \mathbb{R}^{16 \times 16}$ for $i \in [40]$ and we generate $A_i$ as follows, for each row of $A$, we sample from $\mathcal{N}(0, \mathbf{I}_{d \times d})$. This is equivalent to generating each entry of $A$ from Gaussian $\mathcal{N}(0, 1)$. Then, for each data, we select $x \in \mathcal{N}(0, \mathbf{I}_{16})$ and compute $b_i := \langle \exp(A_i^\top x), \mathbf{I}_{16}\rangle^{-1}\exp(A_i^\top x)$.

**Determine the parameter $x$ of softmax regression.**   We select $x$ from $\mathcal{N}(0, \mathbf{I}_{16})$.

**Datasets.**   We generate a validation set for reweight training our model, where it includes 4000 data points. We sample $A_i$ from $\mathcal{N}(0, \mathbf{I}_{16})$, and compute $b_i = \langle \exp(A_i^\top x), \mathbf{I}_{16}\rangle^{-1}\exp(A_i^\top x)$ for $i \in [4000]$. Then we generate test set for testing the performance of algorithms, where it also includes 4000 data points, $A_i$ from $\mathcal{N}(0, \mathbf{I}_{16})$, $b_i = \langle \exp(A_i^\top x), \mathbf{I}_{16}\rangle^{-1}\exp(A_i^\top x)$ for $i \in [4000]$.

**Baselines.**   To evaluate the effectiveness of our approach, we conduct a comparative analysis against two baseline methods referred to as **ICL**, **fine-tuning** and **prefix tuning** Li & Liang (2021). In **ICL**, we provide several input-output examples as the prefix of the model to let them implement in-context learning without any additional training. In **fine-tuning**, we fine-tune the pre-trained model without providing any prefix. These two baselines basically represent the two most common methods of fine-tuning models in real-world cases. In **prefix tuning**, we follow Mangrulkar et al. (2022), concatenate a trainable prefix with embedded input, where the length of the prefix is $16(16+1) = 272$ (same prefix length as RICL).

**Prefixes**    We generate three types of prefixes to evaluate the performance of **ICL**, **RICL**, **LAR-ICL** in executing in context learning under different prefixes. For random prefix, we sample $A_i \in \mathbb{R}^{16 \times 16}$ from $\mathcal{N}(0, \mathbf{I}_{16 \times 16})$, and compute $b_i = \langle \exp(A_i^\top x), \mathbf{I}_{16} \rangle^{-1} \exp(A_i^\top x)$ for $i \in [40]$. For imbalanced prefix, we sample $A_i \in \mathbb{R}^{16 \times 16}$ from $\mathcal{N}(\text{mean}, \mathbf{I}_{16 \times 16})$ and compute $b_i = \langle \exp(A_i^\top x), \mathbf{I}_{16} \rangle^{-1} \exp(A_i^\top x)$ for $i \in [40]$, where mean stands the mean value of distribution. We generate varying degrees of imbalanced prefixes by sampling $A_i$ from distributions $\mathcal{N}(\text{mean}, \mathbf{I}_{16 \times 16})$ with different values of mean. For noisy prefix, we sample $A \in \mathbb{R}^{16 \times 16}$ from $\mathcal{N}(\text{mean}, \mathbf{I}_{16 \times 16})$ and compute $b_i = \langle \exp(A_i^\top x), \mathbf{1}_{16} \rangle^{-1} \exp(A_i^\top x) + \varepsilon$ for $i \in [40]$, where we sample $\varepsilon$ $i.i.d$ from $\mathcal{N}(0, \text{std} \cdot \mathbf{I}_{16})$. We generate vary degrees of noisy prefixes by sampling $\varepsilon$ from distributions $\mathcal{N}(0, \text{std} \cdot \mathbf{I}_{16})$ with different values of std. For imbalanced and noisy prefix, we sample $A_i \in \mathbb{R}^{16 \times 16}$ from $\mathcal{N}(0.4, \mathbf{I}_{16 \times 16})$ and compute $b_i = \langle \exp(A_i^\top x), \mathbf{I}_{16} \rangle^{-1} \exp(A_i^\top x) + \varepsilon_i$ for $i \in [40]$, where we sample $\varepsilon_i$ $i.i.d$ from $\mathcal{N}(0, 0.4\mathbf{I}_{16})$.

**Metrics.**    We use MSE (mean square error) as our metric function, where $\text{MSE}(\widehat{y}, y) = \frac{1}{n} \sum_{i=1}^{n} (y - \widehat{y})^2$. Given a set with $k$ MSE performances $\{\text{MSE}_1, \text{MSE}_2, ..., \text{MSE}_k\}$, we transform it to min-max-scaled $\text{MSE}_i := (\text{MSE}_i - \sigma_{\min})/(\sigma_{\max} - \sigma_{\min})$ for $i \in [k]$, where $\sigma_{\max} := \max\{\text{MSE}_1, \text{MSE}_2, ..., \text{MSE}_k\}$ and $\sigma_{\min} := \min\{\text{MSE}_1, \text{MSE}_2, ..., \text{MSE}_k\}$.

## D.2    THE PERFORMANCES OF REWEIGHT ALGORITHMS ON PREFIXES WITH DIFFERENT DISTRIBUTIONS

We evaluate the performance of ICL, RICL, and LARICL on a test set with three types of prefixes: random, imbalanced, and noisy. Additionally, we assess the performance of fine-tuning by fine-tuning the model without providing any prefix in the input. We record the mean squared error (MSE) for all experiments. Due to the significantly larger MSE of ICL compared to RICL and fine-tuning, we utilize a min-max scaling technique to transform the MSE values into a metric, denoted as "min-max-scaled MSE". A smaller value of "min-max-scaled MSE" indicates a higher MSE and better performance of the algorithm.

## D.3    ROBUSTNESS ON IMBALANCED PREFIX AND NOISY PREFIX

We evaluate the performance of ICL, RICL, and LARICL on a test set with different degrees of imbalanced prefixes and different degrees of noisy prefixes. We set std $= \{0.2, 0.4, 0.6, 0.8, 1.0, 1.2, 1.4, 1.6\}$ as the standard deviation of the distribution of prefix for the left image in Figure 3. We set mean $= \{0.2, 0.4, 0.6, 0.8, 1.0, 1.2, 1.4, 1.6\}$ as the mean value of the distribution of prefix for the right image in Figure 3.