# OpenReview forum: "Fine-tune Language Models to Approximate Unbiased In-context Learning"
_ICLR.cc/2024/Conference — Submitted to ICLR 2024_

### Official Review · Reviewer_NqVg · 2023-10-31

**Soundness:** 2 fair
**Presentation:** 2 fair
**Contribution:** 2 fair
**Rating:** 3
**Confidence:** 3

**Summary:**

This paper uses learning-based approaches to approximate the un-biased in-context learning (ICL). The performance of ICL is highly dependent on quality of demonstrations such as imbalanced or biased demos; this paper investigates the phenomenon in the context of SoftMax self-attention that is built on prior works. Authors also performed experiments to compare their RICL and LARICL algorithms against other fine-tuning and prefix tuning methods on different distribution prefixes.

**Strengths:**

Different from previous in-context learning (ICL) that is focused on linear self-attention (LSA) with linear tasks, based on the pioneering work on ICL for attention scheme (Gao et al 2023), this work focuses on the training process of the attention mechanisms and decompose it to SoftMax regression following Deng et al 2023. Authors attempt to establish that ICL achieves implicit parameter optimization by minimizing the SoftMax regression loss.

**Weaknesses:**

The reviewer concerns on unclear presentation of both methods and experiments. In the former, it is not clear to me what’s their contribution given the cited prior works by Gao et al 2023 and Deng et al 2023. Is it application of the reweighting algorithms on the demonstration examples? In the latter, it is not clear to me about their experimental setup, data set generation, pre-training etc. In Section D of the Appendix, it appears that they assume the input-output demo pairs are composed of a soft prompt with the corresponding output, is it class label or continuous output? What is the real-world task it can correspond to? Change-of-Thought? Using a real-world data set/task with hard prompts can be more meaningful since soft prompt might not make semantic sense when mapping back to hard prompts or cannot even be mapped back to meaningful hard prompts.

**Questions:**

What’s the original contribution of the SoftMax regression objective proposed here compared to Deng et al?

What are the ICL learning tasks you have used in your experiments? If soft prompts have been used, how do you ensure it maps to semantically meaningful hard prompts?

What do you mean by imbalance or bias at all? How does your reweighting approach compare with the simple resampling approaches that simply make the ICL demos balanced and/or unbiased?

---

### Official Review · Reviewer_A8do · 2023-11-04

**Soundness:** 2 fair
**Presentation:** 1 poor
**Contribution:** 2 fair
**Rating:** 3
**Confidence:** 3

**Summary:**

This paper proposes re-weighted in-context learning (ICL) to address the biased and imbalanced input prompts for ICL. The main idea is to fine-tune language models on an unbiased validation set, and to learn the optimal weight for each few-shot example.

**Strengths:**

- The paper points out an interesting problem, i.e., the quality or weight of the prompt / few-shot examples might affect the ICL performance, and it remains an interesting study about how to weight / select the optimal prompts for efficient ICL.

- The paper proposes a neat method of reweighing the embedding vectors of the prompts, where the weight is learned via an unbiased validation set.

- The method is proved promising on some synthetic datasets.

**Weaknesses:**

- I found the paper is super unclear, which makes it difficult to understand the main contribution. Most of the part is devoted to understand the relationship of ICL and softmax regression, which is based on existing work. If I understand it correctly, it basically then studies the re-weighted softmax regression (both theoretically and empirically) instead, to somehow equivalently showing that the study is valid for ICL. Though I appreciate the simplification here, I doubt the applicability of the method on large language models.

- The empirical results are purely based on synthetic numerical datasets, instead of any language models, I am not fully convinced by the empirical results here. It would be great if the author could perform more empirical studies based on existing language models, instead of the "toy examples". Also, it would be great to compare the proposed methods with other PEFT methods on language tasks as well.

- The paper discusses the "unbiased" all the time, but i found even a definition of "unbiasedness" is missing in the paper. What unbiasedness refers to in language models?

- The motivation of the paper is also not well-supported. It would be great to add some understanding on how existing language models assigns weights on various few-shot examples? how the diversity/quality of these examples make a difference for ICL?

**Questions:**

See Weakness.

---

### Official Review · Reviewer_sUUz · 2023-11-21

**Soundness:** 3 good
**Presentation:** 3 good
**Contribution:** 3 good
**Rating:** 5
**Confidence:** 3

**Summary:**

This paper presents RICL and LARICL, algorithms to fine-tune language models to estimate the optimal weights for each in-context example. In-context learning is highly susceptible to the input-output examples, leading to bias and imbalanced learning. This paper addresses this issue by learning parameters implicitly for in-context learning, reweighting the input vectors appropriately to mitigate the biases and imbalances.

**Strengths:**

1. This paper addresses an important problem of ICL being noisy. It proposes learning weights for input vectors to enable unbiased ICL.
2. The authors also extend their algorithm to linear approximation to minimize the training cost of fine-tuning language models.
3. They also study the convergence of their proposed algorithm by first establishing the smoothness of the gradients.

**Weaknesses:**

1. Notation clarification: lot of notations have been used before defining them. For example, on page 2, it is not clear what is R, n and f_i?
Similarly in Theorem 3.4, what does x_t \leq R and x_{t+1} \leq R mean?
2. The bounds presented in theorems seem to be pretty loose, seems like the approximation error will increase with dimension of examples? Also, how does it behave with number of examples?
3. From Fig. 2, prefix-tuning looks like a great contender to proposed algorithms and if I understand correctly, prefix-tuning is less computationally expensive than proposed algorithms.
4. Experiments in the paper are on data generated by the authors. It will be more effective if RICL was able to demonstrate bias mitigation on some public dataset (as simple as few-shot classification tasks).

**Questions:**

1. Just so that I understand better, what if the in-context examples were repeated, A_1=A_2=....A_m. How does RICL would result in more unbiased performance compared to say vanilla ICL? Or does it need some "diversity" in examples?
2. How valid are the assumptions in real-world scenario, especially the ones needed for Lipschitz-smooth gradients

---

### Meta-Review · Area_Chair_xE1h · 2023-12-06

**Metareview:**

The paper proposes a weighted version of in-context learning for fine-tuning LLMs to tasks. All of the reviewers agree that the exposition was unclear, and the connection to softmax regression not sufficiently well-motivated. The lack of any experiments with any LLM was a major weakness, and reviewers concluded that the paper is below the bar for publication.

**Justification For Why Not Higher Score:**

Including experiments with actual LLMs will substantially strengthen the paper. Improving the exposition, e.g. carefully defining bias and the desired goal of unbiasedness in the paper's problem context will also improve the paper.

**Justification For Why Not Lower Score:**

N/A

---

### Decision · Program_Chairs · 2024-01-16

Reject